# Cosmic Tangle: Loop Quantum Cosmology and CMB Anomalies

Martin Bojowald 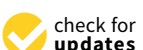

Institute for Gravitation and the Cosmos, The Pennsylvania State University, 104 Davey Lab, University Park, PA 16802, USA; bojowald@gravity.psu.edu

**Abstract:** Loop quantum cosmology is a conflicted field in which exuberant claims of observability coexist with serious objections against the conceptual and physical viability of its current formulations. This contribution presents a non-technical case study of the recent claim that loop quantum cosmology might alleviate anomalies in the observations of the cosmic microwave background.

**Keywords:** loop quantum cosmology; observations



"Speculation is one thing, and as long as it remains speculation, one can understand it; but as soon as speculation takes on the actual form of ritual, one experiences a proper shock of just how foreign and strange this world is." [1]

## 1. Introduction

Quantum cosmology is a largely uncontrolled and speculative attempt to explain the origin of structures that we see in the universe. It is uncontrolled because we do not have a complete and consistent theory of quantum gravity from which cosmological models could be obtained through meaningful restrictions or approximations. It is speculative because we do not have direct observational access to the Planck regime in which it is expected to be relevant.

Nevertheless, quantum cosmology is important because extrapolations of known physics and observations of the expanding universe indicate that matter once had a density as large as the Planck density. Speculation is necessary because it can suggest possible indirect effects that are implied by Planck-scale physics, but manifest themselves on more accessible scales. Speculation is therefore able to guide potential new observations.

Speculation becomes a problem when it is based on assumptions that are unmentioned, poorly justified, or, in some cases, already ruled out. When this happens, speculation is turned into a ritual followed by a group of practitioners who continue to believe in their assumptions and ignore outside criticism. The uncontrolled nature of quantum cosmology makes it particularly susceptible to this danger.

A recent example is the claim [2] that loop quantum cosmology may alleviate various anomalies in observations of the cosmic microwave background. Given the commonly accepted distance between quantum cosmology and observations available at present, this claim, culminating in the statement that "these results illustrate that LQC has matured sufficiently to lead to testable predictions," is surprising and deserves special scrutiny, all the more so because a large number of conceptual and physical shortcomings have been uncovered in loop quantum cosmology over the last few years (all citations from [2] refer to its preprint version).

Upon closer inspection, we encounter a strange world in the claims of [2] according to which, to mention just one obvious misjudgment, "many of the specific technical points [of a recent critique of the methods used in the analysis] were already addressed, e.g., in [42, 64, 65] and in the Appendix of [66]" even though [42, 64, 65, 66] were published between nine and twelve years before the recent critique that they were supposed to have addressed.

Clearly, the authors of [2] have not sufficiently engaged with relevant new criticism. There is therefore a danger that their work is based on rituals. Our analysis will, unfortunately, confirm this suspicion. A dedicated review of these claims is especially important because they have been widely advertized, for instance in a press release. [1]

The present paper provides a close reading of these claims and highlights various shortcomings. It therefore serves as a case study of the complicated interplay between quantum cosmological modeling and observations. A detailed technical discussion of some of the underlying problems of current versions of loop quantum cosmology was already given in [3]. The analysis here is held at a non-technical level and is more broadly accessible. It highlights conceptual problems that are relevant for [2], but were not considered in [3], for instance concerning suitable justifications of initial states in a bouncing cosmological model.

Since the focus of our exposition was on quantum cosmology, other questions that may well be relevant for an assessment of the claims of [2] are not discussed. These questions include a proper analysis of all early-universe observables in addition to a select number of anomalies and the possible role of sub-Planckian unknowns such as further modifications of gravity not considered in [2] or uncertainties about the energy contributions in the early universe; see for instance [4–6] for related examples and reviews.

## 2. Inconsistencies

The model constructed in [2] was based on several key assumptions made elsewhere in the literature on loop quantum cosmology:

**Assumption 1.** *In loop quantum cosmology, there are quantum-geometry corrections that imply a bounce of the universe at about the Planck density;*

**Assumption 2.** *During quantum evolution through the bounce and long before and after, the quantum state of the universe remains sharply peaked as a function of the scale factor;*

**Assumption 3.** *Although the dynamics is modified and geometry is quantum, general covariance in its usual form is preserved on all scales, including the Planck scale;*

**Assumption 4.** *At the bounce, the spatial geometry and matter distribution are as homogeneous and isotropic as possible, restricted only by uncertainty relations for modes of perturbative inhomogeneity.*

These assumptions are necessary for the model to work as it does. For instance, the bounce (Assumption 1) and spatial symmetries (Assumption 4) imply a certain cut-off in the primordial power spectrum that quickly turns out to be useful in the context of anomalies in the cosmic microwave background. A sharply peaked state (Assumption 2) and general covariance (Assumption 3) make it possible to analyze the model by standard methods of spacetime physics, using line elements and well-known results from cosmological perturbation theory [7].

Upon closer inspection, however, it is hard to reconcile these assumptions with established physics. Some of them are even mutually inconsistent. We first discuss these inconsistencies and then examine how the authors tried to justify their assumptions.

### 2.1. Bounce

In general relativity, there are well-known singularity theorems that prove the existence of singularities in the future or the past, under certain assumptions such as energy inequalities or the initial condition of a universe expanding at one time [8,9]. In their most general form, these theorems do not require the specific dynamics of general relativity, but only use the properties of Riemannian geometry, such as the geodesic deviation equation. Loop quantum cosmology can avoid the big bang singularity and replace it by a bounce only if it violates at least one of the assumptions of singularity theorems. Just having a theory with modified dynamics compared with general relativity is not sufficient.

Loop quantum cosmology does not question that the universe is currently expanding, thus obeying the same initial condition commonly used to infer the singular big bang (it is known that the specific modeling of an expanding universe can be weakened so as to evade singularity theorems [10], but loop quantum cosmology does not avail itself of this option). Moreover, as emphasized in [2], "These qualitatively new features arise without having to introduce matter that violates any of the standard energy conditions." According to Assumption 3, spacetime in loop quantum cosmology remains generally covariant on all scales (described by a suitable line element) and therefore obeys the general properties of Riemannian geometry. Loop quantum cosmology does modify the gravitational dynamics, but this is not a required ingredient of singularity theorems.

Since all the conditions of singularity theorems still hold, according to the explicit or implicit assumptions in [2], loop quantum cosmology should be singular just as classical general relativity. How can it exhibit a bounce at high density? For a detailed resolution of this conundrum, see [11].

### 2.2. Peakedness

Setting aside special dynamics such as the harmonic oscillator, quantum states generically spread out and change in complicated ways as they evolve. For a macroscopic object, such as a heavy free particle, it takes longer for such features to become significant than for a microscopic system, but they are nevertheless present.

In late-time cosmology, it is justified to assume that a simple homogeneous and isotropic spatial geometry of a large region describes the dynamics very well. At late times, a state in quantum cosmology may therefore be assumed to remain sharply peaked because it describes a large region that is conceptually analogous to a heavy free particle.

When such a state is extrapolated to the big bang, however, very long time scales are involved. Is it still justified to assume that the state does not change significantly and spread out during these times? Moreover, if the usual assumption of a homogeneous and isotropic geometry is maintained for a tractable description of quantum cosmology, such a region would have to be chosen smaller and smaller as we approach the big bang, not only because of the shrinking space of an expanding universe in time in reverse, but also, and more importantly, because attractive gravity implies structure formation within co-moving volumes. As the big bang is approached, a valid homogeneous approximation of quantum cosmology more and more resembles quantum mechanics of a microscopic object.

Because we do not know what a realistic geometry of our universe at the Planck scale should be, we do not know how macroscopic it may still be considered to be in a homogeneous approximation. The only indication is the Belinskii–Khalatnikov–Lifshitz (BKL) scenario [12] of generic spacetime properties near a spacelike singularity, which suggests that there is no lower limit to the size of homogeneous regions in classical general relativity. Quantum gravity is expected to modify the dynamics of general relativity, but it is not known whether and at what scale it would be able to prevent the BKL-type behavior.

This result suggests that a microscopic description should be assumed in the Planck regime. However, according to [2], "One can now start with a quantum state $\Psi(a, \phi)$ that is peaked on the classical dynamical trajectory at a suitably late time when curvature is low, and evolve it back in time towards the big bang using either the WDW equation or the LQC evolution equation. Interestingly the wave function continues to remain sharply peaked in both cases." Why should quantum cosmological dynamics be such that it maintains a sharply peaked state even over long time scales that include a phase in which the evolved object should be considered microscopic?

### 2.3. Covariance

According to Assumption 1, the bounce in loop quantum cosmology is supposed to happen because quantum geometry modifies the classical dynamics: "Of particular interest is the area gap—the first non-zero eigenvalue $\Delta$ of the area operator. It is a fundamental microscopic parameter of the theory that then governs important macroscopic

phenomena in LQC that lead, e.g., to finite upper bounds for curvature." There is supposed to be a certain quantum structure of space that leads, among other things, to a discrete area spectrum.

However, the description of perturbative inhomogeneity in [2] or the underlyingin [13] assumes that this quantum geometry can be described by a line element. To be sure, co-efficients of the "dressed" metric in this line element contain quantum corrections, but it has not been shown that a quantum geometry with some discrete area spectrum can be described by any line element at all, even at the Planck scale, where discreteness is supposed to be significant enough to change the dynamics of the classical theory. While specific corrections in metric coefficients have been derived, the claim that they should appear in a line element of some Riemannian geometry has not been justified.

There are now several no-go results [14,15] that demonstrate violations of covariance in regimes envisioned by the authors of [2]. A proof that Riemannian geometry and line elements can nevertheless be used in their context would therefore require a detailed discussion of how these no-go results can be circumvented. However, there is no hint of such an attempt. It is worth noting that problems with covariance have also occurred in a different application of the same formalism to black holes, in which several other physical problems were quickly found [16–19]. The spacetime description assumed in [2] is therefore unreliable.

Given the uncertain status of what structure spacetime geometry should have in the presence of modifications from loop quantum cosmology, the meaning of "finite upper bounds for curvature" that are supposed to be implied by the discrete area spectrum is unclear.

### 2.4. Symmetry

Spatial homogeneity and isotropy are assumed at various places in [2]. First, in older papers referred to for justifications of some claims, quantum evolution is numerically computed for states of an exactly homogeneous and isotropic geometry, using methods from quantum cosmology. Secondly, states for inhomogeneous matter perturbations on such a background are assumed to preserve the symmetry as much as possible while obeying uncertainty relations.

### 2.4.1. Background

An attempt was made in [2] to justify these assumptions: "On the issue of simplicity of the LQC description, we note that in the 1980s it was often assumed that the early universe is irregular at all scales and therefore quite far from being as simple as is currently assumed at the onset of inflation. Yet now observations support the premise that the early universe is exceedingly simple in that it is well modeled by a FLRW spacetime with first order cosmological perturbations. Therefore, although a priori one can envisage very complicated quantum geometries, it is far from being clear that they are in fact realized in the Planck regime."

However, this attempted justification is invalid because the scale probed by early-universe observations referred to in this statement is vastly different from the Planck scale in which the outcome is applied. Cosmic inflation was proposed precisely to address, among other things, the homogeneity problem even if the initial state may be much less regular, and inflation is still used in most models of loop quantum cosmology. In such scenarios, large-scale homogeneity at later times does not show in any way that the universe must have been homogeneous at the Planck scale.

The assumed homogeneity is also in conflict with the claimed discrete structure of space that, according to Assumption 1, might imply a bounce. The authors of [2] never addressed the relevant question of how their symmetry assumptions can be reconciled with a discrete geometry on ultraviolet scales that is supposed to imply all the claimed effects (in other words, the authors ignored the trans-Planckian problem of inflationary cosmology [20–22]). Of course, suitable superpositions of discrete states may lead to a

continuum of expectation values even of operators that have a discrete spectrum. However, if this argument were used as a possible explanation of homogeneity, it would put in doubt the strong emphasis on a single eigenvalue $\Delta$ of the area spectrum that is claimed to govern the new dynamics.

### 2.4.2. Perturbations

The authors of [2] used symmetry assumptions not only for the background on which inhomogeneous modes evolve, but also for the modes themselves, described perturbatively. These modes cannot be exactly symmetric because they are subject to uncertainty relations and therefore have, at least, non-zero fluctuations even if their expectation values may be zero. It is claimed that these modes should be as symmetric as possible in the bounce phase, that is have zero expectation values and fluctuations such that they saturate uncertainty relations.

The homogeneity assumption for perturbations is motivated by Penrose's Weyl curvature hypothesis [23], introduced in [2]: "Finally, the principle that determines the quantum state $\Psi(Q, \phi)$ of scalar modes involves a quantum generalization of Penrose's Weyl curvature hypothesis in the Planck regime near the bounce, which physically corresponds to requiring that the state should be 'as isotropic and homogeneous in the Planck regime, as the Heisenberg uncertainty principle allows'." However, conceptually, there is a significant difference between these two proposals. Penrose's hypothesis was given in a big bang setting in which the initial state (close to the big bang singularity) was to be restricted by geometrical considerations. As always, one may question the specific motivation for a certain choice of initial states, but there are no physical objections provided general conditions (such as uncertainty relations) are obeyed.

The symmetry assumption employed for matter perturbations in [2] is of a very different nature. It is used to determine an initial state only of our current expanding phase of the universe, but it is set in a bounce model with a pre-history before the big bang. The symmetry assumption for matter perturbations is therefore a final condition for the collapse phase and violates determinism. Potential violations of deterministic behavior have indeed been derived in models of loop quantum cosmology in the form of signature change at high density [24–26]. However, Reference [2] did not use this option, which would in fact be in conflict with the line element they assumed to formulate a wave equation for cosmological perturbations. Moreover, the derived versions of signature change would set the beginning of the Lorentzian expanding branch of the universe later than assumed in [2], at the very end of the bounce phase in which modifications from loop quantum cosmology subside.

Setting aside the question of determinism, a restriction of the state during the bounce phase, a transitory stage, is not an initial condition, but rather an assumption about the state to which preceding collapse may have led. It is then questionable that gravitational collapse of a preceding inhomogeneous universe should lead to a bounce state that is as homogeneous as possible, respecting uncertainty relations. In this scenario, the collapse of a preceding universe is supposed to have led to a very homogeneous state at the Planck density of more than one trillion solar masses in a proton-sized region. Given the inherently unstable nature of gravitational collapse, one would rather expect that any slightly overdense region in a collapsing universe would quickly become denser and magnify initial inhomogeneities that had been present when collapse commenced.

It is hard to see how collapse could, instead, lead to a state that is as homogeneous as possible. Such an assumption would at least require a dedicated justification, in particular because it directly implies the crucial features of a new scale in loop quantum cosmology that is then used to alleviate anomalies. Unfortunately, no attempted justification can be found in [2].

Upon closer inspection, the arguments given in [2] were even circular. The very setup that led the authors to their formulation of a homogeneous initial condition already assumed the near homogeneity of a collapsing universe: The specific statement in [2] referred to "the Planck regime near the bounce" and implicitly assumed that (in the

Riemannian geometry, according to Assumption 3) there is a time coordinate such that "the bounce" happens everywhere within a large region at the same time. However, if the collapsing geometry is inhomogeneous, overdense regions will become denser during collapse and reach the Planck density earlier than their neighbors. Once they bounce and start expanding, it is not obvious that a simple near-homogeneous slicing with a uniform bounce time still exists (this process suggests a multiverse rather than a single nearly homogeneous universe [27]).

There is no unique bounce time in this picture, and therefore, an implicit assumption (a meaningful "near the bounce") used in the condition of initial homogeneity is unphysical. Crucial statements such as "In our LQC model, the physical principles used to select the background quantum geometry imply that the corresponding ΛCDM universe has undergone approximately 141 e-folds of expansion since the quantum bounce until today." therefore remain unjustified.

### 2.5. Attempted Justifications

The authors of [2] realized that some of these assumptions should be justified, while they were apparently unaware of additional implicit assumptions that they did not mention.

1.　The bounce is justified by quantum-geometry corrections from loop quantum gravity, but the conflict with singularity theorems has apparently gone unnoticed;
2.　The peakedness of states is justified by referring to detailed numerical studies of evolving states in loop quantum cosmology. However, the authors failed to notice that these studies implicitly assume that the universe is still macroscopic (large-scale homogeneity), even in the Planck regime, ignoring structure formation within co-moving regions in a collapsing universe, as well as BKL-type behavior;
3.　Covariance is justified only vaguely by referring to wave equations for perturbations on a background, without asking the relevant question of whether modified perturbation and background equations can still be consistent with a single metric that is being perturbed;
4.　In their discussion of initial conditions for perturbations, the authors seemed to be unaware of problems posed by the pre-history of a collapsing universe.

## 3. A Brief Engagement with the Previous Critique

It is instructive to analyze the brief, but rapid-fire response given in [2] to the previous criticism of certain claims in loop quantum cosmology [3]. As already mentioned in the Introduction, the authors stated that "Many of the specific technical points were already addressed, e.g., in [42, 64, 65] and in the Appendix of [66]." However, the references provided tried (but failed [28,29]) to address an older issue, cosmic forgetfulness [30,31], that did not play a major role in the recent discussion of [3] (since the publication of these older papers, cosmic forgetfulness has been strengthened to signature change).

### 3.1. Effective Descriptions

The authors of [2] went on and stated that "First, although 'effective equations' are often used in LQC, conceptually they are on a very different footing from those used in effective field theories: One does not integrate out the UV modes of cosmological perturbations. The term 'effective' is used in a different sense in LQC: these equations carry some of the leading-order information contained in sharply peaked quantum FLRW geometries $\Psi(a, \phi)$." (Note that the authors were hedging their statement by correctly saying that "these equations carry *some* of the leading-order information" (emphasis added). The fact that they carry only some, but not all of the leading-order information is a problem in itself that will not be discussed here; see [3,32] for details.).

The admission that "the term 'effective' is used in a different sense in LQC" does not make this formalism more meaningful. It is in fact one of the major problems in current realizations of the framework of loop quantum cosmology. Equations of loop quantum cosmology are used on vastly different scales, in the Planck regime to analyze the possibility

of a bounce, and at low curvature to justify a peaked late-time state. A suitable effective theory (not in the sense used, according to [2], in loop quantum cosmology) would be needed to determine how parameters of the model may change by infrared or ultraviolet renormalization. The authors were simply assuming that a single effective theory without any running parameters can be used to describe the quantum universe on a vast range of scales. There is no justification for this assumption.

In addition, the authors referred not only to different scales in cosmology, but also to different geometries, including those of black hole horizons: When they justified certain parameter choices for the dynamics of loop quantum cosmology, they made statements such as "The eigenvalues of [the area operator] $\hat{A}_S$ are discrete in all $\gamma$-sectors. But their numerical values are proportional to [the Barbero–Immirzi parameter (a quantization ambiguity)] $\gamma$ and vary from one $\gamma$ sector to another.", "In LQG, a direct measurement of eigenvalues of geometric operators would determine $\gamma$. But of course such a measurement is far beyond the current technological limits.", and "Specifically, in LQG the number of microstates of a black hole horizon grows exponentially with the area, whence one knows that the entropy is proportional to the horizon area. But the proportionality factor depends on the value of $\gamma$. Therefore if one requires that the leading term in the statistical mechanical entropy of a spherical black hole should be given by the Bekenstein-Hawking formula $S = A/4\ell_{\mathrm{Pl}}^2$, one determines $\gamma$ and thus the LQG sector Nature prefers."

These statements refer to at least three different regimes of some underlying theory of loop quantum gravity: direct microscopic measurements of eigenvalues, macroscopic black hole horizons, and the entire universe at various densities. It is simply assumed that the same value of $\gamma$ (as well as other parameters) may be used in all these situations without suitable renormalization. The only justification attempted for this assumption is the statement that "the term 'effective' is used in a different sense in LQC", which ignores the physical reasons for standard ingredients in effective theories.

The possibility of running is also ignored in statements such as "In IV C, we will show that the interplay between LQC and observations is a 2-way bridge, in that the CMB observations can also be used to constrain the value of the area gap $\Delta$, the most important of fundamental microscopic parameters of LQG.", "As we saw in section III, the area gap $\Delta$ is the key microscopic parameter that determines values of important new, macroscopic observables such as the matter density and the curvature at the bounce. Its specific value, $\Delta = 5.17\ell_{\mathrm{Pl}}^2$, is determined by the statistical mechanical calculation of the black hole entropy in loop quantum gravity (see, e.g., [55, 56, 59, 60]).", and "Clearly, the value $\Delta \approx 5.17\ell_{\mathrm{Pl}}^2$ chosen in Sec. III C and used in this paper, is within 68% ($1\sigma$) confidence level of the constraint obtained from Planck 2018. This not only indicates a synergy between the fundamental theoretical considerations and observational data, but also provides internal consistency of the LQC model."

### 3.2. Covariance

In their reply to [3], the authors stated that "As we will see in Section III B, equations satisfied by the cosmological perturbations are indeed covariant." In Section III.B, however, less than half a sentence was devoted to this important question: "Note also that the equation is covariant w.r.t. [the dressed metric] $\tilde{g}_{ab}$ and $\tilde{g}_{ab}$ rapidly tends to the classical FLRW metric of GR outside the Planck regime." This attempted justification of covariance apparently refers to the equation $(\tilde{\Box} + \tilde{U}/\tilde{a}^2)\hat{Q} = 0$ for modes $\hat{Q}$, with certain functions $\tilde{U}$ and $\tilde{a}$ of the background scale factor. However, nowhere did the authors address the crucial question of whether background $\tilde{g}_{ab}$, which defines the d'Alembertian $\tilde{\Box}$, and perturbation $\hat{Q}$ can, after modifications, still be obtained from a single covariant metric (obeying the tensor-transformation law), as in the underlying classical theory.

The authors' statement about covariance refers only to transformations of the perturbative mode, $\hat{Q}$, and therefore to small inhomogeneous coordinate changes that preserve the perturbative nature. In addition, potentially large homogeneous transformations of the background time coordinate, such as transforming from proper time to conformal time,

are relevant in cosmological models of perturbative inhomogeneity. The usual curvature perturbations are not invariant with respect to these transformations [33]. The authors' statements had nothing to say about the question of whether their model of modified perturbation equations is covariant with respect to large transformations of background time. The failure of covariance in the underlying construction was shown in [14].

*3.3. Symmetry*

We already addressed the authors' erroneous view that observations can be used to justify near homogeneity in the Planck regime. The authors finally stated that "Nonetheless, one should keep in mind that, as in other approaches to quantum cosmology, in LQC the starting point is the symmetry reduced, cosmological sector of GR. Difference from the Wheeler-DeWitt theory is that one follows the same systematic procedure in this sector as one does in full LQG. But the much more difficult and fundamental issue of systematically deriving LQC from full LQG is still open mainly because dynamics of full LQG itself is still a subject of active investigation." Here, they were attempting to construct a false binary choice between exactly isotropic models of symmetry-reduced geometries, on one hand, and calculations in full loop quantum gravity without any symmetry assumptions, on the other. If this choice were correct, one might as well give up because exactly isotropic models are unrealistic, and the full theory is intractable.

What the authors were missing is a proper effective theory that not only amends isotropic equations by certain leading-order corrections, but also tries to go beyond strictly isotropic models by parameterizing all the ignorance in the parameter choices implied by the intractable nature of full loop quantum gravity. Without such an effective description, which does not require direct calculations in the full theory and is therefore feasible, but would not be as simple as the authors assumed, no observational claims can be reliable.

## 4. Conclusions

How can we reconcile the claim that "these results illustrate that LQC has matured sufficiently to lead to testable predictions" [2] with the availability of several serious and independent concerns that have shown in recent years how loop quantum cosmology, as it is commonly practiced, has overlooked a large number of important conceptual and physical requirements? In the present paper, we provided detailed evidence to show that [2] merely ignored or insufficiently addressed relevant criticism. Moreover, the main new claims of [2] are implied rather directly by specific assumptions that remain unjustified. We summarize these observations in this concluding section.

In their quest to show that loop quantum cosmology naturally resolves anomalies in observations of the cosmic microwave background and therefore makes testable predictions, the authors of [2] used or introduced several conceptually distinct assumptions. Some of these assumptions, including those about bounces and covariance, are questionable within loop quantum cosmology. Others, for instance about free parameters, rely on an oversimplified presentation of loop quantum cosmology as some special version of an effective theory that could be used to describe the dynamics of quantum gravity on a vast range of scales, including the Planck regime, without any running parameters.

Yet another set of assumptions, referring to the state of perturbations and their spatial homogeneity, is independent of loop quantum cosmology, but has been packaged with the other assumptions in a way that gives the erroneous impression of a single coherent theory. Moreover, these assumptions are physically questionable because they implicitly make strong and unrealistic claims about the generic final state of a collapsing universe. We are observing a single universe that might perhaps have emerged from a special version of preceding collapse as envisioned by the authors. However, by simply formulating the desired behavior as an assumption without addressing a possible relationship with a pre-history, the authors hid its restrictive nature and ultimately failed to explain the initial state of cosmological perturbations and the observed microwave background.

The authors of [2] were aware of the previous criticism and gave a half-hearted attempt to address it. In Section 3, we saw how inadequate their response was. To summarize, the cited papers that supposedly addressed "many of the specific technical points", made in [3], had been published between nine and twelve years before this recent critique. The crucial question of general covariance in models of quantum gravity, discussed from different viewpoints for instance in [34,35], was acknowledged in [2] only by the misleading "as we will see in Section 3 B, equations satisfied by the cosmological perturbations are indeed covariant" to announce a brief statement "note also that the equation is covariant w.r.t. [the dressed metric] $\tilde{g}_{ab}$" that reflects a misunderstanding of the issue of covariance in the setting of modified perturbation equations (not only perturbations, but also the background must be included in a covariance analysis [14]).

Notably, there are also points of critique that the authors of [2] did not bother to address. An important example, in addition to the prevalence of quantization ambiguities, is the implicit assumption in current formulations of loop quantum cosmology that the universe remains large-scale homogeneous (over at least several hundred Planck lengths) even at the Planck density. As explained in [3], this assumption is related to the incorrect implementation of effective descriptions in loop quantum cosmology. Without this unjustified and unmentioned assumption, the authors of [2] would not even be able to formulate their restrictive condition that cosmological perturbations be as homogeneous as possible at the bounce. This assumption ultimately leads to a suppression of power on large scales of the cosmic microwave background and allows claims about resolved anomalies. However, this assumption is not only unjustified, but it is also based on another and implicit assumption that had already been ruled out.

Therefore, the results of [2] relied on several major assumptions, made explicitly or implicitly, that turned out to be unjustified. The authors' claims therefore went well beyond the usual level of speculation that is common (and unavoidable) in quantum cosmology. Their response to objections that have been published during the last few years was inadequate in some cases and non-existent in others. Ashtekar et al. created a cosmic tangle of rituals that they no longer wish to be questioned.

**Funding:** This research was funded by NSF Grant Number PHY-1912168.

**Conflicts of Interest:** The author declares no conflict of interest.

## Note

1    https://news.psu.edu/story/626795/2020/07/29/research/cosmic-tango-between-very-small-and-very-large (accessed on 2 June 2021).

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
