# Peer review of "Cosmic Tangle: Loop Quantum Cosmology and CMB Anomalies"

_universe, doi:10.3390/universe7060186_

Round 1

Reviewer 1 Report

It's a well written article that criticises another article through what appear to be sound arguments. It also clarifies some of assumptions made in loop quantum cosmology. I recommend publication. 

Author Response

Thank you very much for this positive review.

Reviewer 2 Report

In his manuscript "Cosmic tangle: Loop quantum cosmology and CMB anomalies" Prof. Bojowald provides a critical examination of very recent speculations of his colleagues from the Institute of Gravitation and the Cosmos at PSU concerning possible observable signatures of Loop Quantum Cosmology (LQC) on CMB anomalies. As one of the world leading experts in Loop Quantum Gravity (LQG) and LQC the author has a first-hand intuition about the basic assumptions and possible loopholes of the forming theory complex of LQG/LQC. Especially, this concerns possibly or certainly too speculative assumptions in the setups as used in the various research schools of his community.  From this point of view, his critical examinations in the present manuscript and a previous one (ref. [3])  appear very timely and important. What seems missing in the critical examination (and what might be of interest to colleagues from outside the LQG/LQC community) would be well founded comments on possible additional imprints on CMB anomalies from interaction aspects of high-energy/high-density matter components on intermediate energy scales still a bit away from the Planck scale where LQG/LQC effects show up. Intuitively, these colleagues might be tended to suspect that such matter-interaction effects on intermediate energy scales might lead to an even stronger imprint on CMB anomalies than LQG/LQC effects --- masking them additionally. A clarifying comment on the interplay of the imprints on the various energy scales would be helpful for such colleagues from outside the LQG/LQC community --- just to avoid not necessary questions, and even if these aspects appear obvious to the author.

Anyway, the only verdict concerning the present manuscript: immediate quick publication (in its present form) in "Universe" --- also to allow for a substantial external input into the probably still ongoing review process of the criticized manuscript [2].

Optionally and as subject for a future publication, it would be highly welcome to see the author's critical examinations extended to similarly speculative concepts from competing theory complexes outside LQG/LQC such as the various string-theoretic/M-theoretic concepts on physical setups close to big-bang and black-hole singularities --- although these topics are outside of the author's direct research field. 

In summary: immediate publication of the present manuscript in its present form in "Universe". Optionally the author might extend his critical evaluation of [2] by comments on interacting high-energy/high-density matter at intermediate energy scales a bit away from the assumed LQG/LQC scales. 

Author Response

Thank you very much for the review and suggestions. I have appended a
paragraph to the Introduction in which I mention the issues of sub-Planckian effects and possible implications of unknown matter/energy contributions in the early universe. Since most of these questions are outside my area of expertise, I refer to several recent reviews. 

An extension of my critique to non-loop approaches would also be outside my core area of expertise. Moreover, the specific arguments used in my paper rely to some degree on the fact that loop quantum cosmology is a canonical approach which faces unique challenges. These arguments are therefore not immediately extendable to other approaches, but this might certainly be a possible future endeavor.

Reviewer 3 Report

I have examined  this manuscript, in which the author  addresses the Loop quantum cosmology (LQC) approach and tries to emphasize its (extra)limitations. He admits right from the start that it  is a conflicted field in which claims of observability  coexist with  objections against the  viability of its current formulations. The author focus his criticisms on recent works in which claims are made that LQC can solve a variety of tensions currently afflicting the standard LCDM (or concordance) model of cosmology.  The manuscript is written in a descriptive form and avoids going into too technical details,  it is what the author calls a non-technical case study of the recent claim that  LQC  could  alleviate the mentioned anomalies. The author refers specifically to Ref. [2], where strong claims are made on the resolution of the aforementioned anomalies if using the LQC approach.  In particular, it is claimed that LQC has a clue on why the observed TT-power spectrum  is suppressed relative to the theoretical prediction for low multipoles (typically \ell< 30).  They also focus on the lensing amplitude anomaly A_L, which is set to 1 in the standard CMB analysis under the LCDM, but generates  conflict with observations.  The authors, however, do not address  the discrepancy between the results of the SHOES team and CMB measurements associated with the value of the Hubble parameter, nor on the trouble in the structure formation data, namely the tension  in between measurements of the amplitude of the power spectrum of density perturbations inferred using CMB data against those directly measured by Large-Scale Structure formation on smaller scales, from redshift space distortions.  From this point of view the purported capacity of LQC to fix all these tensions is at the moment rather limited.  Not only so, with a simpler framework (such as e.g. Brans-Dicke gravity with a cosmological term) it is possible to highly alleviate all these tensions at a time, see e.g.    arXiv:2006.04273.  In particular,   the presumed clue possessing LQC as to why the observed TT-power spectrum  is suppressed relative to the theoretical prediction is also accounted for in  the last paper (see Fig. 12)  and with no A_L anomaly at all.  At the same time both H_0 and sigma_8  increase and diminish, respectively, and the residual tensions exhibited by these observables are rendered at the level of 1.5\sigma or less.  In general, dynamical models of the vacuum energy help along this same direction, as recently pointed out in  some references.  So no special need for LQC and the associated plethora of sophisticated assumptions underlying this framework if one just wants to face practical issues of current cosmology such as the mentioned tensions. The current author knows well the `inside story' about LQC and in this descriptive work summarizes a lists of pros and cons related with this story,  in particular the cons that could make difficult to implement it, such as the concerns on whether the general physics principles of effective field theory and covariance have been consistently incorporated in such framework. A previous critical account on these difficulties was already presented by the author in Ref. [3].   The main thesis of the author was that at present, it is not possible to derive a sufficiently complete effective theory from full loop quantum gravity, and even if this were possible, it is not clear whether LQG itself is covariant and consistent.  The answer by the authors of [2] to that were that these issues had been answered in previous papers.  The current work seems not to agree and it abounds further along these lines and in particular it criticizes the possibility that LQC is sufficiently free of conflicting issues as to apply it reliably to two particular tensions (as those mentioned above) and infer from the obtained (partial) alleviation a kind of phenomenological accolade to reinforce its status.  In particular, I agree with the author that some of the statements made in the press release cited on p. 2  look rather  exaggerated.  For example, when one of the authors involved in the LQC studies  mentions that "Using quantum loop cosmology, however, we have resolved two of these anomalies naturally, avoiding that potential crisis".  This is just preposterous.  Even if they have been "resolved" it is not obvious what are the potentially weird consequences for other observables.  As indicated above, there is nothing special in alleviating these two tensions unless the other mentioned two are also (simultaneously) alleviated, as in the case of the mentioned studies, which could be of interest for the readers as well. From this point of view I think that the debate on LQC remains open and the work presented by the author may contribute to motivate further studies on its theoretical consistency and its ability to solve the persistent discrepancies of the LCDM with different sources of cosmological data.  Therefore, I can essentially recommend this manuscript for publication in Universe.  I would only recommend the author to smooth out some statements which do not add anything substantial in his (manifest) discrepancies with the authors of Ref. [2].  After all, the only thing that matters is improving  the scientific content of the paper, nothing else.

Author Response

Thank you very much for your review and the additional discussion and suggestions. I have appended a paragraph to the Introduction in which I mention possible other problems that I, for lack of direct expertise, do not discuss in my paper. The same paragraph refers to relevant papers in this direction, including the one about Brans-Dicke gravity suggested in the review. 

I had already made an attempt to write in objective language before I first submitted the paper, but a few unnecessary phrases have remained. I weakened them further by, in particular, changing "they have contrived a set of" to "their work is based on" (page 2), removing "supposed to be" (page 4), removing "completely" (preceding "ignore," page 5) and "highly" (preceding "questionable," page 5), changing "supposed to" to "then used to" (page 5), and removing two other "completely" (preceding "unaware," page 6, and preceding "ignores," page 7).